# PI3K/mTOR Inhibitor Induces Context-Dependent Apoptosis and Methuosis in Cancer Cells

**DOI:** 10.3390/ph18121849

**Published:** 2025-12-04

**Authors:** Xiaoyuan Hua, Panpan Chen, Wanjing Zeng, Yuqiao Han, Yanzhi Guo, Yanmei Chen, Chuchu Li, Yijie Du, Mingliang Ma, Suzhen Dong

**Affiliations:** 1Shanghai Engineering Research Center of Molecular Therapeutics and New Drug Development, School of Chemistry and Molecular Engineering, East China Normal University, Shanghai 200062, China; huaxiaoyuan0@163.com (X.H.); chenpanpan2024@163.com (P.C.); 51264300151@stu.xmu.edu.cn (W.Z.); yqhan1995@163.com (Y.H.); russgyz@foxmail.com (Y.G.); 18130355565@163.com (Y.C.); 32320230157391@stu.xmu.edu.cn (C.L.); 2Qingpu Traditional Chinese Medicine Hospital, Shanghai 201799, China; 3Key Laboratory of Brain Functional Genomics-Ministry of Education, School of Life Science, East China Normal University, Shanghai 200062, China

**Keywords:** YYN-37, PI3K/mTOR inhibitor, methuosis, apoptosis, VPS34

## Abstract

**Background/Objectives:** Targeting the PI3K/mTOR pathway is a promising strategy in cancer therapy, but its efficacy is often limited by apoptosis resistance. This study investigates the dual PI3K/mTOR inhibitor YYN-37, exploring its capacity to induce context-dependent cell death, particularly the non-apoptotic process of methuosis. **Methods:** We examined the effects of YYN-37 on HCT-116 and SJSA-1 cancer cell lines using cell viability assays, Western blot, and fluorescent tracers. Cell death mechanisms were probed with pathway-specific inhibitors. The role of VPS34 was assessed through kinase activity assays, siRNA-mediated knockdown, and rescue experiments with the agonist leucine. Proteomic profiling and an in vivo SJSA-1 xenograft model in BALB/c nude mice were utilized to evaluate broader mechanisms and anti-tumor efficacy. **Results:** YYN-37 induced caspase-3-dependent apoptosis in HCT-116 cells. In contrast, it triggered a reversible, cytoplasmic vacuolization in SJSA-1 cells, identified as methuosis. This vacuolization originated from endocytic pathways and was inhibited by EIPA and Baf-A1. YYN-37 directly inhibited VPS34 (IC_50_ = 2.73 nM), and its knockdown replicated the vacuolization, which was conversely reversed by the VPS34 agonist leucine, confirming VPS34-dependency. Proteomics revealed lysosomal dysfunction in SJSA-1 cells and cell cycle alterations in HCT-116 cells. In vivo, YYN-37 treatment resulted in a 72.71% tumor growth inhibition, with histology confirming methuosis-like vacuolization. **Conclusions:** YYN-37 exerts potent, context-dependent anti-tumor effects by inducing apoptosis in HCT-116 cells and VPS34-mediated methuosis in SJSA-1 cells. This work establishes methuosis induction as a viable therapeutic strategy for apoptosis-resistant cancers and highlights VPS34 inhibition as a promising mechanism of action.

## 1. Introduction

The PI3K/mTOR pathway is a crucial intracellular signaling pathway involved in modulating a multitude of biological processes, including cell growth, proliferation, survival, and metabolism [1]. Among the three classes of PI3K members identified, class I PI3Ks have been most extensively studied. These class I enzymes, subdivided into α, β, γ, and δ isoforms based on their regulatory subunit composition, specifically catalyze the transformation of phosphatidylinositol 4,5-diphosphate (PIP2) into phosphatidylinositol 3,4,5-triphosphate (PIP3), a key signaling event in this pathway. Subsequently, PIP3 activates mTOR via its effector molecule AKT. mTOR serves as a sensor for external environmental factors such as growth factors, energy levels, and nutrient availability, subsequently regulating cellular physiological functions. Notably, class III PI3K (homologous to yeast PIK3C3/Vps34) is involved in regulation of autophagy and endocytic trafficking by phosphorylating phosphatidylinositol (PI) to generate phosphatidylinositol 3-phosphate (PI3P) [2]. Dysregulation of this pathway has been strongly implicated in the development of numerous human cancers, thus positioning it as a prime target for anti-cancer drug development [3]. A number of inhibitors targeting this pathway have been employed in cancer therapeutics [4]. Over 40 such inhibitors have advanced to various phases of clinical trials, with 9 having secured clinical approval [5].

Research has considered the PI3K/AKT signaling pathway as a pivotal modulator in cell death processes. It has been demonstrated to be closely linked with a spectrum of cell death types, including apoptosis, necroptosis, autophagic cell death, pyroptosis, ferroptosis, and cuproptosis [6,7]. Inhibitors targeting the PI3K/mTOR pathway have been shown to combat tumors by triggering apoptosis or autophagy in cancer cells [8,9,10,11,12,13]. Notably, nitidine chloride has been found to induce caspase 3/GSDME-dependent pyroptosis in lung cancer by suppressing the PI3K/Akt pathway [14]. Furthermore, compounds eliciting autophagic cell death concurrently induce dephosphorylation of mammalian target of rapamycin complex 1 (mTORC1) [15]. Emerging studies have also revealed that aspirin blocks the AKT/mTOR signaling cascade, thereby fostering ferroptosis in colorectal cancer cells [16]. Inhibition of the cannabinoid receptor type 1 has been observed to lead to ferroptosis in a PI3K-dependent manner within triple-negative breast cancer cells [17]. The dual-targeting PI3K and HDAC inhibitor BEBT-908 facilitates ferroptosis in cancer cells by hyperacetylating p53 and amplifying the expression of ferroptosis-associated signals [18]. Sensitivity to the PI3K inhibitor TGX221 correlates positively with the expression levels of COX7B or SLC25A5, genes implicated in cuproptosis, in esophageal carcinoma [19]. These findings imply that the PI3K/AKT/mTOR pathway participates in a variety of cell death forms, and its potential involvement in additional newly identified cell death types is worth further investigation. Overall, more research is needed to fully understand how the PI3K/AKT pathway works in different cells, which depends on cell type, context, and/or stimulus.

Our group had previously developed a dual-target PI3K/mTOR inhibitor YYN-37, characterized by notable anti-tumor activities both in vitro and in vivo, good kinase selectivity, minimal hepatotoxic effects, moderate plasma clearance rates, and satisfactory oral bioavailability [20]. It is worth highlighting that this inhibitor elicited disparate death responses across distinct tumor cell lines. Specifically, HCT-116 cells mainly underwent programmed cell death (apoptosis), while SJSA-1 cells showed cytoplasmic vacuolization, suggesting that the compound might work differently depending on the cell type. Previous studies showed that vacuolization-associated death include paraptosis, autophagic cell death, and methuosis [21]. Therefore, we further investigated the cell death mechanisms induced by the compound in SJSA-1 cells.

Based on our findings, we propose that the compound may induce methuosis, a distinct non-apoptotic cell death characterized by cytoplasmic vacuolization caused by dysregulated endocytic/vesicular trafficking pathways [22]. Contrary to apoptosis, methuosis does not lead to DNA fragmentation or caspase activation; instead, it culminates in cell death via metabolic disintegration and cell membrane rupture. Research indicates a strong correlation between methuosis and the dysfunction of the endosomal-lysosomal pathway, with VPS34 serving as a pivotal regulatory element of this pathway. By forming complexes with binding proteins such as Rab5 and UVRAG (UV radiation resistance-associated gene), VPS34 specifically controls a range of key cellular processes, including the fusion of endosomes and lysosomes, the maturation of autophagosomes, and the dynamic regulation of vesicular transport [23]. Dysfunction of VPS34 has been implicated in neurodegenerative diseases, metabolic disorders, and a spectrum of cancers [2], its precise role in tumor cell death remains unclear.

To investigate anti-tumor mechanisms of the compound and to elucidate connections between PI3K/mTOR signaling and distinct cell death pathways, we undertook a comprehensive analysis of YYN-37′s effects in HCT-116 and SJSA-1 cells, and performed a preliminary exploration of its underlying mechanisms. This study was aimed at uncovering the anti-tumor mechanisms of YYN-37 across different cellular contexts, thereby offering foundational insights and innovative perspectives for the development of anti-cancer therapeutics targeting the PI3K/mTOR pathway.

## 2. Results

### 2.1. Compound YYN-37 Induces Different Forms of Cell Death in Various Tumor Cells

In our team’s previous research, compound YYN-37 exhibited robust anti-tumor activity both in vitro and in vivo. Utilizing a xenograft model of human colon cancer cells in nude mice, YYN-37 markedly inhibited tumor growth. Moreover, it reduced the levels of phosphorylated AKT and p70S6K in a dose- and time-dependent manner, indicating that YYN-37′s anti-tumor efficacy is mediated through the suppression of PI3K and mTOR kinase activities [20].

Interestingly, while we selected SJSA-1 cell line (osteosarcoma) to evaluate whether YYN-37, whose effects had been well-documented in colorectal cancer models (HCT116) [20], could also inhibit osteosarcoma cells for which effective therapies are scarce, we unexpectedly observed strikingly differential responses between the two cell types (Figure 1A). Following a 24 h treatment with YYN-37, HCT-116 cells displayed characteristics of cell rounding, shrinkage, and budding; in contrast, SJSA-1 cells showed pronounced cytoplasmic vacuolization without signs of shrinkage or budding. Subsequent testing in cell lines such as HT29, HOS, MNNG/HOS, and MDA-MB-231 revealed that YYN-37 could induce significant cytoplasmic vacuolization in HOS, MNNG/HOS, A549 and MDA-MB-231 cells (Appendix A). This suggests that YYN-37 is capable of eliciting distinct morphological alterations across various cell types. Given the characteristic reactions of HCT-116 and SJSA-1 cells, we selected these two cell lines for further in-depth investigation.

First, we analyzed whether the most common form of cell death, apoptosis, occurred in the two cell types. After treating HCT-116 and SJSA-1 cells with different concentrations of YYN-37 for 24 h, we used Western blot to detect the cleavage of caspase-3. The results showed that in HCT-116 cells, the protein level of cleaved-caspase-3 increased with the concentration of YYN-37 after treatment, indicating that apoptosis occurred in these cells; however, the truncated form of caspase-3 was undetectable in SJSA-1 cells, indicating no significant induction of apoptosis (Figure 1B,C). In further experiments, HCT-116 or SJSA-1 cells were treated with YYN-37 in the presence or absence of the apoptosis inhibitor Z-VAD-FMK, and then cell viability was analyzed using the CCK-8 assay. The results showed that Z-VAD-FMK significantly reversed the inhibitory effect of YYN-37 on the viability of HCT-116 cells, but had no significant effect in SJSA-1 cells, supporting the notion that YYN-37 induces apoptosis in HCT-116 cells, but not in SJSA-1 cells (Figure 1D,E). We further verified the different effects of YYN-37 in the two cell types through TUNEL staining. After treatment with YYN-37, HCT-116 cells showed significant green fluorescence, indicating DNA fragmentation and apoptosis occurred in the cells; however, almost no fluorescence signal was observed in SJSA-1 cells, confirming that apoptosis did not occur in SJSA-1 cells (Figure 1F). Subsequently, an investigation was conducted to determine if YYN-37 caused autophagy within the two cellular variants. As shown in Figure 1G,H, 3-methyladenine (3-MA), an inhibitor of autophagy, exerted no discernible effect on YYN-37′s suppression of cellular proliferation, thereby indicating an absence of significant autophagy induction by YYN-37 in either cell line. In addition, electron microscopy results showed that YYN-37 induced numerous single-membrane vacuoles, rather than double-membrane autophagosomes, in SJSA-1 cells (Appendix A). These results revealed that YYN-37 predominantly induced cell death via apoptosis in HCT-116 cells. Conversely, YYN-37 failed to induce apoptosis in SJSA-1 cells and instead elicited cytoplasmic vacuolization, hinting at the potential activation of a non-apoptotic programmed cell death form. According to literary references, it was hypothesized that this phenomenon might be attributable to methuosis, a cell death process caused by disrupted vesicular trafficking.

### 2.2. YYN-37 Induces Cytoplasmic Vacuolization

Firstly, morphological changes were analyzed in detail in SJSA-1 cells after YYN-37 treatment. Vacuoles started to form after just one hour of treatment, with their size and number increasing as time and concentration progressed, peaking between 6 and 12 h. By 24 to 48 h, the vacuoles had enlarged and began to merge, ultimately rupturing and causing cell death. Consequently, YYN-37 triggers a time- and concentration-dependent cytoplasmic vacuolization in SJSA-1 cells (Figure 2A). To identify the derivation of vacuoles, a comparison was made between the vacuoles induced by YYN-37 and those induced by the known PI3K inhibitors related to vacuoles, SAR405 [24] and Apilimod [25] (Figure 2B). The findings revealed that all three compounds could induce vacuole formation, but to different degrees. Notably, the vacuoles induced by YYN-37 looked more like those induced by SAR405, characterized by a more scattered distribution and variable sizes, whereas the vacuoles induced by Apilimod were more compact and uniform in size. Furthermore, the reversibility of the vacuolization process induced by YYN-37 was examined. Upon removal of the compounds after 48 h, the vacuoles in the SAR405 and YYN-37 groups progressively diminished and the cells resumed normal proliferation. In contrast, the vacuoles in the Apilimod group remained unchanged even 24 h after compound withdrawal (Figure 2C). Thus, it can be concluded that the vacuolization process induced by SAR405 and YYN-37 is reversible, suggesting a similar mechanism of formation, while the process induced by Apilimod appears to differ.

### 2.3. The Cytoplasmic Vacuolization Induced by YYN-37 Can Be Reversed by Inhibitors of Methuosis

In order to determine if the vacuolization induced by YYN-37 is associated with methuosis, we employed the recognized methuosis inhibitors EIPA and Baf-A1 [22,26] to assess their potential to reverse this effect. Our findings revealed that the vacuolization triggered by YYN-37 was markedly inhibited following a 1 h pre-treatment with either 25 µM EIPA or 100 nM Baf-A1 (Figure 2D). This outcome implies that the cytoplasmic vacuolization elicited by YYN-37 could be indicative of methuosis. Nevertheless, an alternative methuosis inhibitor, Filipin [27], a cholesterol binding agent, did not suppress the YYN-37-induced vacuolization at concentrations of either 5 µM or 20 µM (Figure 2E). This suggests that the mechanism by which YYN-37 induces vacuole formation is not connected to cholesterol trafficking and distribution. Overall, the ability of EIPA and Baf-A1 to block vacuole maturation points towards the possibility that the vacuolization induced by YYN-37 comes from disruptions in the cellular vesicular transport mechanisms.

### 2.4. The Cytoplasmic Vacuolization Induced by YYN-37 Is Associated with Endocytosis

Paraptosis is another form of non-apoptotic cell death characterized by cytoplasmic vacuolization, with vacuoles deriving from mitochondria and the endoplasmic reticulum [28]. To elucidate the origin of the vacuoles induced by YYN-37, we employed fluorescent tracers ER-Tracker Green and Mito-Tracker Red to label the endoplasmic reticulum and mitochondria in SJSA-1 cells for live cell staining. Intriguingly, the vacuoles induced by YYN-37 did not colocalize with either tracer (Figure 3A,B), suggesting that these vacuoles did not arise from the mitochondria or endoplasmic reticulum, and thus YYN-37-induced cell death was not paraptosis. Using the natural acidification properties of endocytic vesicles, we performed fluorescence colocalization experiments with acridine orange (AO), a metachromatic dye that specifically accumulates in acidic compartments. In normal cells, AO staining exhibits green fluorescence in the nucleus and red fluorescence in a small number of acidic vesicles. The experimental results indicated that after 24 h of YYN-37 treatment, the vacuoles in SJSA-1 cells colocalized with the red fluorescence of AO, indicating that these vacuoles were acidic vesicles associated with the endocytic pathway (Figure 3C).

Further, a large molecular fluorescent dye Dextran-FITC was used to label endocytosis. After YYN-37 treatment, some vacuoles were observed to colocalize with fluorescence, confirming its association with endocytosis and macropinocytosis (Figure 3D). Additionally, Lucifer Yellow was used to further track the endocytic pathway. Cells treated with YYN-37 and SAR405 showed a significant increase in intracellular fluorescence, indicating that these two compounds may promote endocytosis or hinder excretion, thereby increasing the accumulation of the dye within the cells (Figure 3E). The results suggest that the vacuolization induced by YYN-37 may be closely related to the intracellular vesicular transport pathway.

### 2.5. The Effects of Compound YYN-37 on Methuosis-Related Death Pathways

The literature indicates that methuosis is associated with disruption of the endosome-lysosome pathway and disorder of vesicular transport, usually accompanied by changes in the activity of small GTPases [22]. To further confirm YYN-37 induces methuosis in SJSA-1 cells, we examined the expression changes in early endosome marker Rab5A, late endosome marker Rab7A, and lysosome marker Lamp1 after 24 h of treatment with different concentrations of YYN-37. The results showed that with increasing concentrations of YYN-37, the expression of Rab7A and Lamp1 increased, while the level of Rab5A decreased at high concentrations (Figure 3F). This suggests that YYN-37 may inhibit the normal fusion process between late endosomes and lysosomes, leading to the accumulation of lysosomal and late endosomal components within the cells, and may thereby interfere with cellular degradation and recycling functions. Moreover, small GTPase Rac1 and GTP-binding protein Arf6 have been shown to be regulated when methuosis occur [29,30,31]. Our results also showed that YYN-37 treatment raised Rac1 protein levels but lowered Arf6 expression (Figure 3F). This imbalance likely disrupts cellular endocytosis and vesicle transport systems, thereby further inhibiting the fusion process between endosomes and lysosomes.

To further investigate whether YYN-37 inhibits the fusion of late endosomes with lysosomes, we examined the impact of YYN-37 on the maturation status of cathepsin D (CTSD), a protease that undergoes processing via the endocytic pathway during its transport into lysosomes. The results showed that after treatment with YYN-37, the levels of prepro-CTSD and pro-CTSD increased, but the level of mature CTSD decreased (Figure 3G). This indicates that YYN-37 may inhibit the maturation of CTSD during the transport from endosomes to lysosomes. Collectively, these results suggest that YYN-37 may affect lysosomal function by interfering with the normal fusion of endosomes and lysosomes, thereby leading to cellular dysfunction.

### 2.6. YYN-37-Induced Vacuole Formation Is Dependent on VPS34

YYN-37 has been demonstrated to exert potent inhibition across all PI3K isoforms and mTOR kinases while maintaining high selectivity [20]. Notably, YYN-37 induces cytoplasmic vacuoles morphologically comparable to those generated by SAR405 (Figure 2B), a well-characterized highly potent VPS34 inhibitor [24], suggesting potential VPS34 involvement in its mechanism. To investigate this hypothesis, we employed AO fluorescence staining to examine whether leucine, a known Vps34 agonist [32], could counteract YYN-37-induced vacuolation in SJSA-1 cells. The results revealed that the number of red-fluorescent AO-stained vacuoles induced by YYN-37 treatment decreases with the addition of leucine dose-dependently, with complete vacuole elimination achieved at 2 μM leucine (Figure 4A and Appendix A). These observations imply YYN-37-mediated vacuolation is VPS34-dependent, prompting us to evaluate its direct inhibitory activity against VPS34. Kinase inhibition assays demonstrated potent VPS34 targeting with an IC_50_ of 2.73 nM (Figure 4B). To further validate the mechanistic role of VPS34, we conducted siRNA-mediated VPS34 knockdown in SJSA-1 and A549 cell lines. Remarkably, VPS34 silencing caused cytoplasmic vacuolization in both cell lines (Figure 4C,D and Appendix A), providing conclusive evidence that YYN-37-induced vacuole formation is mediated via VPS34-dependent pathways.

### 2.7. Proteomics Analysis of the Possible Underlying Mechanisms

To investigate YYN-37-induced cellular death mechanisms, we performed proteomic analyses on SJSA-1 and HCT-116 cell lines. SJSA-1 cells were treated with 10 µM YYN-37 and HCT-116 cells with 1 µM YYN-37 for 24 h, followed by comprehensive proteomic profiling.

In SJSA-1 cells, hierarchical clustering revealed distinct protein expression patterns between treatment groups (Figure 5A), with scatter plot analysis identifying 799 differentially expressed proteins (DEPs: 347 upregulated, 452 downregulated; fold change = 1.5, *p* < 0.05; Figure 5B). Subsequent GO enrichment analysis demonstrated significant association of DEPs with vacuolar components, particularly “lysosomal lumen”, “vacuole membrane”, and “lytic vacuole” (Figure 5C). The KEGG pathway GSEA analysis showed significant inhibition of lysosomal function pathways (Figure 5D). This coordinated dysregulation of lysosome-related cellular components and pathways strongly suggests YYN-37 disrupts endolysosomal trafficking homeostasis in SJSA-1 cells.

Notably, HCT-116 cells exhibit fundamentally different response patterns. Proteomic analysis showed predominant enrichment in cell cycle-related GO terms, particularly “cell cycle process” and “mitotic cell cycle process” (Appendix A). This implies YYN-37 may induce cell cycle arrest or apoptotic pathways to inhibit proliferation in HCT-116 cells, a finding consistent with previous cellular experimental results.

These findings collectively demonstrate cell type-specific mechanisms of YYN-37 action: while primarily targeting endolysosomal trafficking in SJSA-1 cells, it appears to predominantly affect cell cycle regulation in HCT-116 cells. This mechanistic divergence underscores the compound’s context-dependent biological effects.

### 2.8. In Vivo Anti-Tumor Activity of Compound YYN-37 in Osteosarcoma Model

Our previous studies have shown that YYN-37 has excellent anti-tumor effects in HCT-116 xenograft nude mice models [20]. We further investigated its therapeutic potential in apoptosis-resistant SJSA-1 osteosarcoma-bearing mice. The SJSA-1 xenograft model received daily oral YYN-37 administration, yielding significant therapeutic outcomes. YYN-37 treatment markedly inhibited tumor volume expansion compared to vehicle controls (Figure 6A), achieving 72.71% tumor growth inhibition (TGI) based on final tumor weights (Figure 6C,D). No significant body weight alterations were observed throughout the treatment course. The transient weight variation on final measurement day lost statistical significance when accounting for tumor mass subtraction (Figure 6B, *p* > 0.05). Histopathological examination confirmed YYN-37-induced vacuolization mirroring in vitro observations (Figure 6E), suggesting similar methuosis-like cell death mechanisms in this animal model. These results collectively demonstrate YYN-37′s robust in vivo anti-tumor activity in osteosarcoma models independent of apoptosis induction, with histopathological evidence supporting vacuolization-mediated cell death as a probable mechanism. The concordance between in vitro and in vivo vacuolization phenomena strengthens the hypothesis of methuosis involvement in the therapeutic effects of YYN-37.

## 3. Discussion

Our findings demonstrate that YYN-37 induces distinct cell death mechanisms in different tumor cell lines, highlighting the context-dependent nature of PI3K/mTOR pathway modulation. In HCT116 cells, YYN-37 triggered apoptosis through caspase-3 activation, a mechanism consistent with previous observations of PI3K/mTOR inhibitors [12,20]. However, in SJSA-1 cells, YYN-37 induced single-membrane cytoplasmic vacuolization without caspase activation or DNA fragmentation, suggesting a non-apoptotic cell death mechanism. This divergence in cell death pathways underscores the complexity of PI3K/mTOR signaling regulation across different cellular contexts and genetic backgrounds.

We found that YYN-37 induced dose-dependent reversible single-membrane cytoplasmic vacuolization in SJSA-1 cells. Meanwhile, the vacuoles were acidic and colocalized with endocytic markers, indicating their derivation from endocytic pathways rather than mitochondrial or endoplasmic reticulum sources. Moreover, pretreatment with methuosis inhibitors EIPA and Baf-A1 significantly reduced vacuolization, while the cholesterol-binding agent Filipin had no effect, suggesting a mechanism distinct from other vacuolization-associated cell deaths like paraptosis. In addition, YYN-37 treatment altered the expression of endocytic markers (Rab5A, Rab7A, Lamp1) and disrupted the maturation of cathepsin D, indicating impaired endosome-lysosome fusion and lysosomal function. All these results support that the YYN-37-induced vacuolization in SJSA-1 cells exhibited characteristics consistent with methuosis, a non-apoptotic cell death form resulting from hyperactivation of macropinocytosis and disrupted endocytic vesicular trafficking [22,33]. Therefore, YYN-37 can be a potential inducer of methuosis in specific cellular contexts, expanding the therapeutic possibilities of PI3K/mTOR inhibitors beyond apoptosis-centric approaches.

Our study identified VPS34 as a critical mediator of YYN-37-induced vacuolization in SJSA-1 cells. YYN-37 exhibited potent inhibition of VPS34 activity (IC_50_ = 2.73 nM) and triggered vacuolization patterns morphologically similar to those induced by the established VPS34 inhibitor SAR405 [24]. Like SAR405, which disrupts PI3P-dependent membrane trafficking and impairs late endosome-lysosome fusion, leading to cytoplasmic vacuolation, lysosomal dysfunction, and defective cathepsin D maturation, YYN-37 also induced reversible, single-membrane vacuoles derived from endocytic compartments. This mechanism is consistent with prior findings demonstrating that loss of VPS34 kinase activity leads to a spatially restricted depletion of PI3P, particularly in late endosomes, without affecting early endosomal PI3P pools or the localization of PI3P-binding proteins such as EEA1. This specific PI3P loss disrupts late endosome maturation and fusion with lysosomes, rather than causing a global loss of PI3P [34]. A similar spatially restricted mechanism may underlie the vacuolization observed in our experiments following YYN-37 treatment.

However, while SAR405 has been primarily characterized as an autophagy and vesicle trafficking inhibitor, YYN-37 represents a first-in-class dual PI3K/mTOR inhibitor that induces methuosis, a non-apoptotic cell death mechanism, in SJSA-1 cells. Notably, leucine, a known VPS34 agonist, dose-dependently reversed YYN-37–induced vacuolization, further supporting the compound’s mechanistic link to VPS34. Genetic knockdown of VPS34 in both SJSA-1 and A549 cells led to spontaneous vacuolization, consistent with prior reports [34], conclusively establishing central role of VPS34 in this process. These findings position VPS34 as a key regulatory node in the mechanism of action of YYN-37 and highlight its potential as a therapeutic target for apoptosis-resistant cancers. While VPS34 knockdown has previously been associated with cytoplasmic vacuolization in U251 cells [34], its connection to methuosis remained unclear. Our results now provide the first evidence implicating VPS34 in the regulation of methuosis, though the precise mechanistic details require further exploration.

In addition, our findings demonstrate cell type-specific mechanisms underlying YYN-37′s action: while HCT116 cells undergo classical apoptosis, SJSA-1 cells develop VPS34-associated methuotic vacuolization. This mechanistic divergence suggests different engagement of PI3K isoforms: type I PI3K inhibition likely drives apoptotic pathways in HCT116 cells, whereas type III PI3K (VPS34) modulation appears responsible for methuosis induction in SJSA-1 cells. The observed VPS34-dependent vacuolization pattern, confirmed through acridine orange colocalization assays, indicates distinct pathway activation between cell types. These results propose novel isoform-selective targeting properties of YYN-37, though further investigation is required to validate PI3K binding specificity, determine cellular determinants of target preference, and characterize downstream signaling interactions.

Proteomic profiling uncovered divergent cellular responses to YYN-37 treatment that align with observed variations in cell death types. In SJSA-1 cells, the compound caused significant dysregulation of endolysosomal trafficking, evidenced by pronounced enrichment of lysosomal components (e.g., CTSD). Conversely, HCT-116 cells displayed predominant alterations in cell cycle checkpoints and mitotic regulation, with key regulators like CDK2 and PCNA showing marked expression changes. This mechanistic dichotomy demonstrates YYN-37′s context-dependent targeting priorities, likely deriving from lineage-specific variations in baseline signaling networks (such as genetic alterations or aberrant expression of PI3K/mTOR pathway members) and compensatory adaptation capacities. While these findings underscore the critical importance of tumor biological context in therapeutic response, the precise molecular drivers of this selectivity require systematic investigation using multi-omics approaches. The demonstrated heterogeneity directly informs drug development pipelines, advocating for biomarker-driven patient stratification strategies that integrate proteomic signatures with genomic landscapes to optimize clinical outcomes.

The in vivo efficacy of YYN-37 in an osteosarcoma model that it does not cause apoptosis demonstrates its potential as a broad-spectrum anti-cancer agent. When used at the dose of 20 mg/kg, YYN-37 showed 72.71% tumor growth inhibition and no significant toxicity. The histopathological confirmation of vacuolization in tumor tissues supports the methuosis mechanism observed in vitro. These results suggest that YYN-37 may offer clinical benefits in cancers where apoptosis pathways are compromised or resistant, potentially expanding the therapeutic options for difficult-to-treat malignancies.

While this study elucidates key mechanisms of YYN-37 action, several questions remain. The specific molecular determinants governing whether cells undergo apoptosis or methuosis in response to PI3K/mTOR inhibition require further investigation. Additionally, the potential for combination therapies targeting both apoptotic and non-apoptotic pathways should be explored to maximize therapeutic efficacy. In addition to regulation of autophagy and vesical trafficking, VPS34 has recently been demonstrated to be involved in immune recognition and tumor micro-environment interactions [35]. Thus, Our VPS34 inhibitor YYN-37 holds promise for broader therapeutic applications and warrants further investigation. Finally, the translation of these findings into clinical settings will require biomarker development to identify patients most likely to benefit from YYN-37 treatment based on their tumor’s biological characteristics.

## 4. Materials and Methods

### 4.1. Reagents

YYN-37 is compound **42** in our published article [20], and was synthesized in our lab according to this study. The structure and detailed characterization data of the compound YYN-37 are provided in the Appendix A).

The ER-Tracker Green fluorescent staining agent, Mito-Tracker Red fluorescent staining agent, Lyso-Tracker Red fluorescent staining agent and CCK-8 cell proliferation assay kit were all purchased from Beyotime (Shanghai, China). Z-VAD-FMK, SAR405, Apilimod, EIPA, Baf-A1, and AO were all obtained from Yuanye (Shanghai, China).

### 4.2. Cells

HCT-116, SJSA-1, and all other cells were purchased from Cell bank of China Academy of Sciences (Shanghai, China), and cultured aseptically in 5% CO_2_ at 37 °C with 1640 or McCoy’s 5A media (Basal media Technologies Co., Ltd., Shanghai, China) supplemented with 10% (*V*/*V*) fetal bovine serum (BI) and 100 units per mL each of penicillin G and streptomycin (Basal media Technologies Co., Ltd., Shanghai, China).

### 4.3. Animals

Female BALB/c nude mice were obtained from Shanghai JIHUI Laboratory Animal Co., Ltd. (Shanghai, China) and maintained under specific pathogen-free (SPF) conditions. All experimental procedures strictly complied with China’s Laboratory Animal Management Regulation and the Three Rs principles (Replacement, Reduction, Refinement) established by William Russell and Rex Burch, with ethical approval granted by East China Normal University.

### 4.4. Cell Viability Assay

Cells were seeded in 96-well plates. After allowing the cells to adhere to the wells and recover their morphology, the culture medium was replaced with the drug-containing medium. After an appropriate incubation period, the cell viability was determined using the CCK-8 assay kit as previously described [36].

### 4.5. Western Blotting

Cells were treated as described, collected and lysed with RIPA buffer (CWBIO, Shanghai, China) containing PMSF (1 mM, Beyotime, Shanghai, China). Protein concentration was measured by the BCA Protein Assay Kit (Kingmorn, Shanghai, China). Western blotting was performed the same as previously described [36]. Briefly, Equal-volume samples (containing 20 μg of protein) were separated by SDS—PAGE and then transferred to PVDF membranes. After blocking with 5% non—fat milk powder at room temperature for 2 h, the membranes were incubated with the specified primary antibodies overnight at 4 °C, followed by incubation with appropriate secondary antibodies conjugated to horseradish peroxidase for 1 h. Enhanced chemiluminescence was used to visualize the immunoreactive bands. The molecular weight of the detected proteins was determined by comparison with pre-stained protein markers. The β-actin, Rab5A, Rab7A, LAMP1, and Cathepsin D primary antibodies were purchased from Proteintech Group, Inc. (Shanghai, China). The secondary antibodies, including Goat anti-Rabbit IgG (H+L) and Goat anti-Mouse IgG (H+L), were obtained from Cell Signaling Technology (Boston, MA, USA).

### 4.6. TUNEL Assay

Cells from treatment, positive control (pre-treated with DNase I to induce DNA breaks), and negative control groups were fixed in 4% paraformaldehyde (10–15 min), permeabilized with—0.1% Triton X-100 (10 min), and incubated with TUNEL reagent (Beyotime Biotechnology) at 37 °C for 1 h. After PBS washes (3 × 5 min), TUNEL-positive apoptotic cells were observed via fluorescence microscopy.

### 4.7. VPS34 Knockdown by RNAi

VPS34 knockdown in SJSA-1 and A549 cells was achieved by siRNA transfection using Lipofectamine™ 6000 transfection kit (Beyotime) according to the manufacturer’s direction. VPS34 siRNA (Sense: 5′-GTGTGATGATAAGGAATAT-3′, Antisense: 5′-ATATTCCTTATCACAC-3′) and its stable non-specific siRNA (negative control) were provided by Gemma Biotechnology (Shanghai, China). After transfection, the knockdown efficiency of VPS34 was verified by Western Blotting.

### 4.8. Proteomic Labeling and Quantification

TMT/iTRAQ—labeled proteomic analysis of cell lysis samples from YYN37-treated HCT-116 and SJSA-1 cells and control cells were conducted in Majorbio Proteomic Service (Shanghai, China) following the manufacturer’s guidelines.

### 4.9. VPS34 Inhibitory Activity Assay

The inhibitory activity of YYN-37 against hVPS34 was evaluated using the ADP-Glo luminescent assay (performed by Shanghai ChemPartner Co., Ltd., Shanghai, China). The compound was diluted in DMSO, transferred to a 384-well plate, and mixed with 2× enzyme solution, substrate, and ATP in kinase buffer. After incubation at 28 °C, ADP-Glo Reagents 1 and 2 were added sequentially to terminate the reaction. Luminescence (measured via PerkinElmer Envision) was used to calculate inhibition percentages, and IC_50_ values were derived using XLFit (v5.4.08) with a four-parameter logistic model.

### 4.10. In Vivo Anti-Tumoral Assay

SJSA-1 cells were cultured, harvested, and suspended in PBS (5 × 10^6^ cells/0.1 mL) for subcutaneous inoculation into BALB/c nude mice. When tumors reached about 100 mm^3^, mice were randomized into two groups (n = 6): treatment (YYN-37, 20 mg/kg orally in 5% DMSO/45% PEG400/50% ddH_2_O) and vehicle control. Tumor volume (V = L × W^2^/2) and body weight were monitored every 3 days. At endpoint, tumors were removed and weighed. Tumor growth inhibition (TGI) was calculated as [1 − (TW_treat_/TW_veh_)] × 100. Statistical analysis used GraphPad Prism 10.0 (one-way ANOVA, * *p* < 0.05).

### 4.11. H&E Staining

Fixed tumor tissues (4% paraformaldehyde) were processed by Servicebio Technology (Wuhan, China). Paraffin sections underwent sequential dewaxing (eco-friendly solution: 2 × 20 min; absolute ethanol: 2 × 5 min; 75% alcohol: 5 min), while thawed frozen sections were fixed (15 min). Staining included hematoxylin (3–5 min) with differentiation/bluing, 95% ethanol dehydration (1 min), and eosin (15 s). After ethanol/xylene treatment, samples were mounted in neutral balsam and imaged using an Olympus IX-73 microscope (Olympus Corporation, Tokyo, Japan).

## 5. Conclusions

The dual-target PI3K/mTOR inhibitor YYN-37 drives context-dependent cell death: apoptosis in HCT-116 cells via caspase-3 activation and methuosis in SJSA-1 cells via VPS34 inhibition (IC_50_ = 2.73 nM), which disrupts endolysosomal trafficking. In vivo, YYN-37 suppressed osteosarcoma growth by 72.71%, with vacuolization mirroring methuosis. This study highlights methuosis as a critical anti-cancer mechanism for apoptosis-resistant tumors and establishes VPS34 as a therapeutic target. The findings emphasize tumor-context-driven strategies to optimize PI3K pathway inhibitors and advance precision oncology paradigms.

## Figures and Tables

**Figure 1 pharmaceuticals-18-01849-f001:**
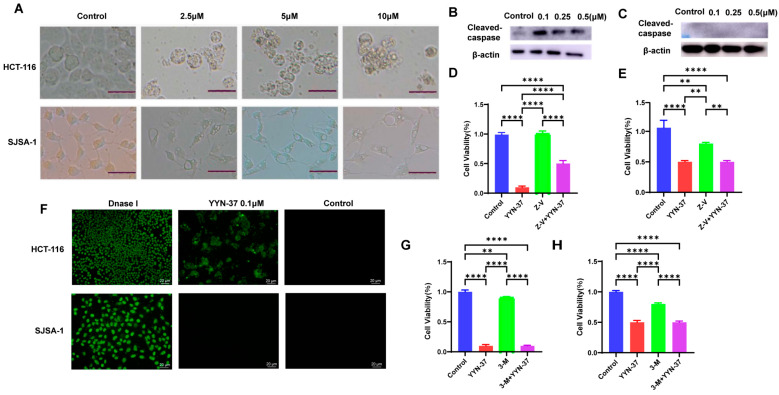
YYN-37 induces apoptosis in HCT-116 cells, but not in SJSA-1 cells. (**A**): Morphological changes in HCT-116 and SJSA-1 cells after 24 h treatment with compound YYN-37. Scale bar = 25 μm (**B**,**C**): HCT-116 (**B**) and SJSA-1 (**C**) cells were treated with YYN-37 at the indicated concentration for 24 h. Then the levels of cleaved caspase 3 were analyzed by Western blot. (**D**,**E**): HCT-116 (**D**) and SJSA-1 (**E**) cells were treated with YYN-37 alone or in combination with the apoptosis inhibitor Z-VAD-FMK(Z-V). After 24 h, cell activity was analyzed by CCK-8. (**F**): HCT-116 and SJSA-1 cells were treated with 0.1 μM YYN-37 for 24 h and analyzed by TUNEL staining. DNase I-treated cells were used as positive control. (**G**,**H**): HCT-116 (**G**) and SJSA-1 (**H**) cells were treated with YYN-37 alone or in combination with the autophagy inhibitor 3-MA(3-M). After 24 h, cell activity was analyzed by CCK-8. Results are expressed as the mean ± SEM (*n* = 3 for each group) and analyzed by one-way ANOVA analysis: ** *p* < 0.01, **** *p* < 0.001.

**Figure 2 pharmaceuticals-18-01849-f002:**
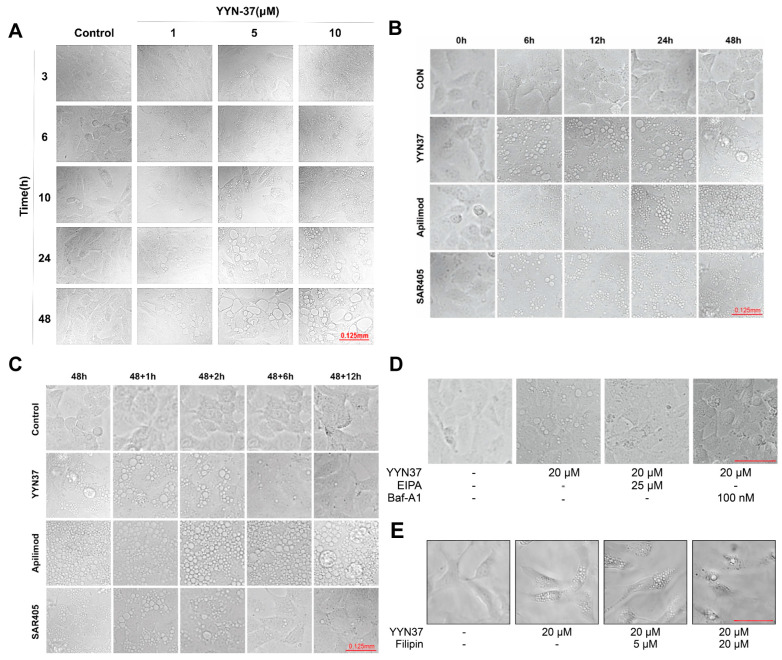
YYN-37 induces reversible cytoplasmic vacuolization and its reversal by methuosis inhibitors. (**A**): The morphological changes in SJSA-1 cells treated with the indicated concentration of YYN-37 for different time. (**B**): The morphological changes in SJSA-1 cells over time after being treated with 20 μM YYN-37, 50 μM SAR405, or 20 μM Apilimod, respectively. (**C**): The morphological changes in SJSA-1 cells over time after the removal of the compounds, which had been used to treat the cells for 48 h. YYN-37 (20 μM), SAR405 (50 μM), or Apilimod (20 μM). (**D**): Pre-treatment with compounds EIPA and Baf-A1 can reverse the vacuolization induced by YYN-37 in SJSA-1 cells. (**E**): Filipin pre-treatment cannot reverse the vacuolization induced by YYN-37 in SJSA-1 cells. Scale bar in D and E = 25 μm.

**Figure 3 pharmaceuticals-18-01849-f003:**
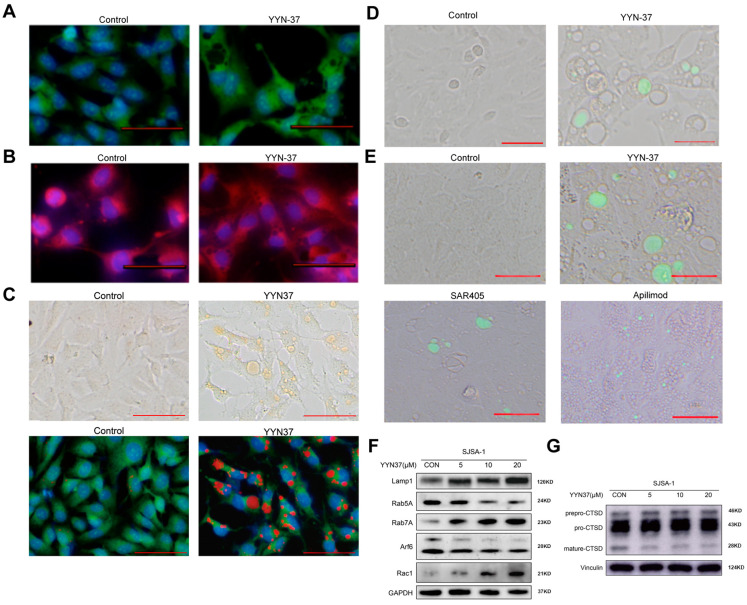
YYN-37-induced cytoplasmic vacuolization is associated with endocytosis. (**A**–**C**): SJSA-1 cells were treated with 20 μM YYN-37 for 1 h and stained with ER-Tracker Green ((**A**) green: endoplasmic reticulum), Mito-Tracker Red ((**B**) Red: mitochondria), or AO ((**C**) red: acidic organelles). The blue fluorescence in A-C is stained by Hoechst 33342 to show cell nuclei. (**D**): SJSA-1 cells were pre-treated with 1 mg/mL Dextran-FITC and then 20 μM YYN-37 or vehicle was added. The uptake of the dye was shown. (**E**): SJSA-1 cells were pre-treated with 5 mg/mL Lucifer Yellow. Then vehicle, compound YYN-37 (20 µM), SAR405 (50 µM), and Apilimod (20 µM) were added. The cells’ uptake of the dye was shown. (**F**,**G**): SJSA-1 cells were exposed to indicated concentrations of YYN-37 for 24 h, followed by a Western blot analysis of total cellular extracts with antibodies against endosome-related proteins (**F**) or CTSD (**G**). The experiments were performed in three replicates. All the scale bars are equal to 25 μm.

**Figure 4 pharmaceuticals-18-01849-f004:**
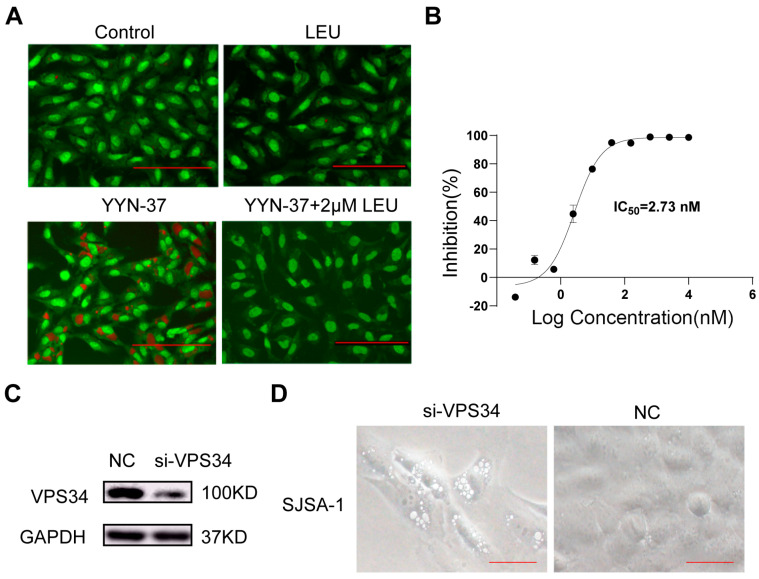
YYN-37-induced vacuole formation is VPS34-dependent. (**A**): SJSA-1 cells were treated with 5 μM YYN-37 in the presence or absence of leucine (2 μM) and stained with AO. green fluorescence: nuclei/cytoplasm), red fluorescence: acidic vacuoles. (**B**): YYN-37 shows potent inhibition on VPS34 activity with an IC_50_ of 2.73 nM. (**C**,**D**): SJSA-1 cells were transfected with VPS34 siRNA or scrambled siRNA. Then VPS34 expression was analyzed by Western blot (**C**). The morphology of the cells was shown in (**D**). Scale bar = 25 μm.

**Figure 5 pharmaceuticals-18-01849-f005:**
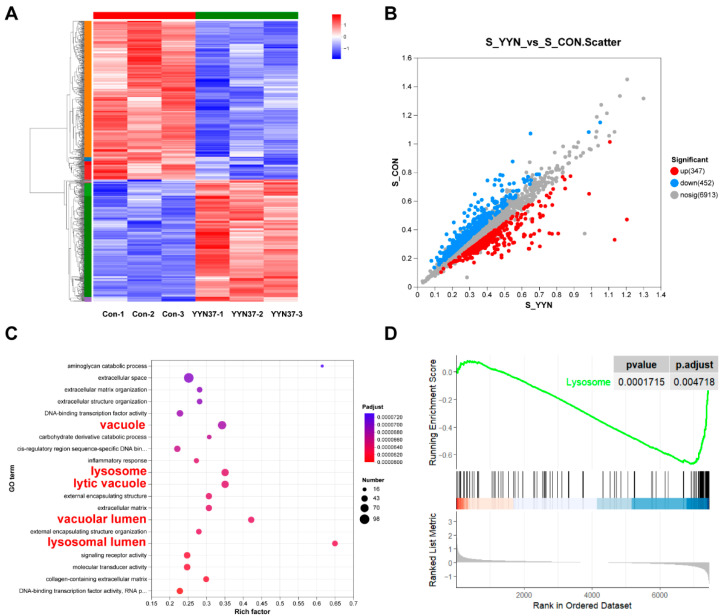
Proteomic Analysis revealed that YYN-37 treatment disrupted lysosome function in SJSA-1 cells. (**A**): Cluster analysis diagram of differential proteins between SJSA-1 cells treated with YYN-37 and the control cells. (**B**): Scatter plot of differential proteins (**C**): GO enrichment analysis map. (**D**): GSEA enrichment analysis plot.

**Figure 6 pharmaceuticals-18-01849-f006:**
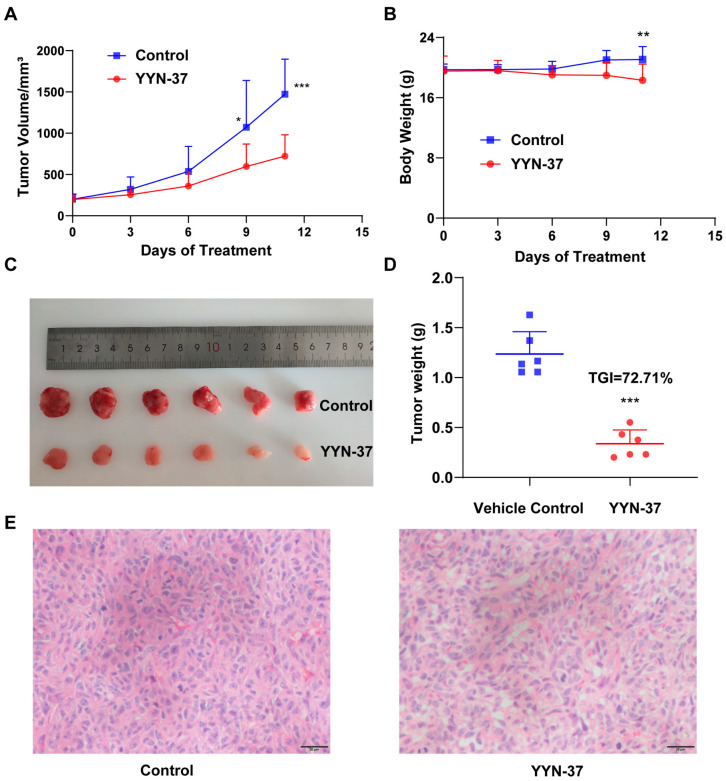
YYN-37 inhibited tumor growth of SJSA-1 xenograft in nude mice. (**A**): Tumor volume changes following daily treatment of YYN-37 at a dose of 20 mg/kg (p.o.) or vehicle beginning on the 7th day after inoculation. (**B**): Average body weight of the mice in control group and YYN-37-treated group. (**C**): Tumors removed from YYN-37- and vehicle-treated mouse groups. (**D**): Tumor weight of YYN-37- and vehicle-treated groups. (**E**): HE staining of tumor tissues from YYN-37-treated mice and vehicle groups. Scale bar = 20 μm. Results are expressed as the mean ± SEM (*n* = 6 for each group) and analyzed by one-way ANOVA analysis: * *p* < 0.05 vs. vehicle, ** *p* < 0.01, *** *p* < 0.001.

## Data Availability

The original contributions presented in this study are included in the article/Appendix A. Further inquiries can be directed to the corresponding authors.

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
