# Peer review of "PI3K/mTOR Inhibitor Induces Context-Dependent Apoptosis and Methuosis in Cancer Cells"

_pharmaceuticals, 2025, doi:10.3390/ph18121849_

Round 1
Reviewer 1 Report
Comments and Suggestions for Authors
The manuscript by Hua et al. presents an investigation into the dual-target PI3K/mTOR inhibitor YYN-37, demonstrating its ability to induce context-dependent cell death via apoptosis in HCT-116 cells and methuosis in SJSA-1 cells. The study is well-designed, the data are robust, and the findings are significant, as they elucidate a non-apoptotic cell death mechanism that could be leveraged for treating apoptosis-resistant cancers. The identification of VPS34 as a critical mediator of the methuotic response is a particular strength. The manuscript is generally well-written, and the experimental evidence supports the conclusions. However, several points require clarification to strengthen the manuscript further.
Major Points
1.The authors convincingly show that YYN-37 inhibits VPS34 and that VPS34 knockdown phenocopies the vacuolization. However, the manuscript would be significantly strengthened by a more direct exploration of how the inhibition of this Class III PI3K leads to methuosis. Is it solely through the loss of VPS34 kinase activity and subsequent reduction in PI3P, disrupting endolysosomal trafficking? The proteomic data shows inhibition of lysosomal function, but a more targeted analysis of PI3P levels or the localization of PI3P-binding proteins (e.g., EEA1, WIPI) upon YYN-37 treatment could solidify this central mechanism.
2.The most interesting finding is the cell-type-specific death mechanism. The authors speculate this may be due to "lineage-specific variations in baseline signaling networks (particularly PTEN status and PI3K isoform expression profiles)." This is a critical point that remains largely hypothetical. The authors should provide data on the baseline status of key nodes in the two cell lines. At a minimum, this should include: i). PTEN status (mutation, expression level); ii). Expression levels of Class I PI3K isoforms and VPS34; and iii). Basal activation status of AKT and mTOR. Furthermore, to test the hypothesis directly, it would be informative to see if sensitizing HCT-116 cells to methuosis is possible (e.g., by knocking down VPS34 in combination with YYN-37) or if forcing a pro-apoptotic state in SJSA-1 cells (e.g., with a BCL-2 inhibitor) shifts the balance. This would provide profound insight into the molecular switches governing the death decision.
3.Several key findings, particularly in the microscopy-based figures (e.g., Figures 1A, F; 2A-E; 3A-E; 4A, D, F), rely on representative images. The conclusions would be much stronger with quantitative data. Provide quantification for: The percentage of cells with vacuoles in Figure 2; The fluorescence intensity or area of colocalization in Figure 3.; The number of TUNEL-positive cells in Figure 1F; and the percentage of cells with vacuoles after VPS34 knockdown in Figure 4D and F. Statistical analysis should be clearly reported for all quantitative experiments, including the number of replicates (n) and the statistical test used. Some bar graphs (e.g., Figure 1D-E, G-H) lack indicators of statistical significance.
4.The authors use inhibitors (EIPA, Baf-A1) to link vacuolization to methuosis, which is good evidence. However, to more definitively characterize the cell death as methuosis, a key experiment is missing: demonstrating that the vacuoles originate from macropinosomes. A direct assay for macropinocytosis, such as tracking the uptake of high-molecular-weight (70 kDa) dextran, should be performed. The current Dextran-FITC and Lucifer Yellow data (Figure 3D-E) suggest endocytic accumulation but do not specifically distinguish macropinocytosis from other endocytic pathways. Showing that YYN-37 stimulates fluid-phase uptake would strongly support the methuosis claim.
Minor Points
The introduction could be more focused by presenting the extensive list of cell death types linked to PI3K/AKT (lines 6-7 on Page 2) to improve flow.
The discussion would benefit from a paragraph explicitly comparing and contrasting YYN-37 with other known VPS34 inhibitors like SAR405 and PIK-III, highlighting the novelty of its methuosis-inducing capability in this specific context.
In Figure 1, the labels (A, B, C, etc.) in the legend do not perfectly align with the sub-panels described. Please review and correct for clarity.
Figure 3F-G and Figure 4C, E: Western blot data should include molecular weight markers. The description "The experiments were performed in three replicates" is good, but it would be standard to show all three replicates in the supplement or a representative blot from one experiment with the others in the supplement.
The term "cytoplasmic vacuolization" is used throughout. After it is established to be associated with endosomes/lysosomes, consider using more specific terms like "endocytic vacuolization" or "lysosome-derived vacuolization" where appropriate.
Page 4, line 5: "SJSA-1" is misspelled as "SISA-1".
Page 9, Figure 4C and E: The text "western blo�ing" appears to be a formatting error and should be corrected to "western blotting."
Reviewer 2 Report
Comments and Suggestions for Authors
In this manuscript, Hua et al. build on previous work with an mTOR inhibitor YYN-37. They show that it induces apoptosis in HCT-116 cells and cytoplasmic vacuolization in SJSA-1 cells. Significantly, YYN-37 inhibited the growth of SJSA-1 tumor xenografts in mice. The data support the claims of the manuscript. However, I have two comments for revision:
- The authors will need to provide descriptions of the cell lines, HCT-116 and SJSA-1, early in the narrative to give readers a sense of their origin and the rationale behind their selection for this study. (Section 2.1)
- I found Figure 4 somewhat confusing: Why did the authors include cell line A549? What is the significance of this cell line in the context of the study? (Figures 4E & 4F).
Minor:
- Figure 4B: Concentration is misspelled.
Reviewer 3 Report
Comments and Suggestions for Authors
The manuscript reported the in vitro studies of effects of YYN-37 in an osteosarcoma model but the title did not clearly show the intents of the present study. How would the nude animal model help physiologically in animal models with different cancers? The results could not demonstrate its potential as an anti-cancer agent, without sufficient data on toxicity assessment. There are voluminous studies on anti-cancer agents in vitro and in vivo studies. How would the present study contribute to the drug development with YYN-37 and its analogues? Alternative approaches have to be conducted to provide supportive evidence on cell cycle and inhibition of cancer metastasis. In addition, what is the significance of results in Figure 6D?
Comments on the Quality of English LanguageNeeds some editing
Round 2
Reviewer 1 Report
Comments and Suggestions for Authors
No further comments
Reviewer 3 Report
Comments and Suggestions for Authors
The manuscript has been sufficiently revised and warranted publication.
Comments on the Quality of English LanguageNeeds some editing